# Global scale analysis on the extent of river channel belts

Björn Nyberg [1,2] ✉, Gijs Henstra[3], Rob L. Gawthorpe [1], Rodmar Ravnås[3] & Juha Ahokas[3]

Rivers form channel belts that encompass the area of the river channel and its associated levees, bars, splays and overbank landforms. The channel belt is critical for understanding the physical river evolution through time, predicting river behavior and management of freshwater resources. To date, there is no global-scale, quantitative study of the extent of river channel belts. Here we show, based on a pattern recognition algorithm, the global surface area of channel belts at an approximate 1 km resolution is $30.5 \times 10^5 \, km^2$, seven times larger than the extent of river channels. We find 52% of river channels associated with the channel belts have a multi-threaded planform with the remaining 48% being single-threaded by surface area. The global channel belt (GCB) datasets provide new methods for high-resolution global scale landform classifications and for incorporating the channel belt into flood mitigation, freshwater budgets, ecosystem accounting and biogeochemical analyses.

Rivers are widely recognized as an essential part for life on Earth supporting ecosystems[1,2], influencing our climate[3] and providing freshwater resources[4]. Rivers can also be destructive with an estimated 1 billion people living in flood-prone regions causing an annual projected 1250 billion Euros in socio-economic damage by the year 2050[5]. The planform character of a river and its channel belt provides an archive of the past river evolution and information on the expected behavior of the river system in the future[6]. With river flooding events expected to increase in both intensity and frequency during this century due to climate change[5], knowledge of the type of river system is vital for flood adaptation strategies[6]. The channel belt environment also supports unique riverine ecosystems and contribute to biogeochemical cycles of carbon that remain understudied at a global scale[2,7]. Once buried, channel belt deposits become part of the geological record and may form subsurface reservoirs that are important for $CO_2$ sequestration[8] and as freshwater aquifers[4].

The river channel belt is defined as the corridor of river channel migration formed during one river avulsion cycle[9] (Fig. 1). The planform characteristics that defines the channel belt extent includes: (1) the active river channel itself and associated bars that are actively accreting and/or migrating; (2) the immediate overbank with levees and/or lateral splays, and (3) channel reaches (and associated bars,

levees and splays) that were abandoned not by nodal avulsions, but by subordinate events such as meander cut-offs[10]. Floodplain material within the channel-belt may include backswamp landforms deposited between the bars, levees or splays associated with the active or abandoned river channels. The planform of the active river channel(s) may be single-threaded channels or multi-threaded channels with an overall straight, sinuous, meandering, braided or anabranching river channel morphology[11,12]. The bars associated with the active river channel include point bars accreting on the inner bank of meandering river channels, mid-channel (braid) bars formed in middle of a river channel and lateral (side) bars attached to the riverbank.

In recent years, integration of data from satellite missions has allowed compilation of detailed high-resolution global scale studies of landcover and water surface change[13–16]. Yet, previous global scale classifications of river systems are either limited to classifications of drainage networks[2,17], descriptions of river morphologies based on high-resolution imagery[18,19], or geometrical measurements of the river channel belt based on a relatively small selection of manually interpreted river systems[20]. Only recently have Allen and Pavelsky[3] calculated that the global surface area of rivers at a 30 m² resolution covers an estimated 468,000 km² or 0.35% of Earth's non-glaciated land surface. This work has also been expanded to show the historical change

[1]Department of Earth Sciences, University of Bergen, Allegaten 41, 5020 Bergen, Norway. [2]Bjerknes Centre for Climate Research, Allegaten 70, 5020 Bergen, Norway. [3]AkerBP ASA, Oksenøyveien 10, 1366 Lysaker, Norway. ✉e-mail: bjorn.nyberg@uib.no

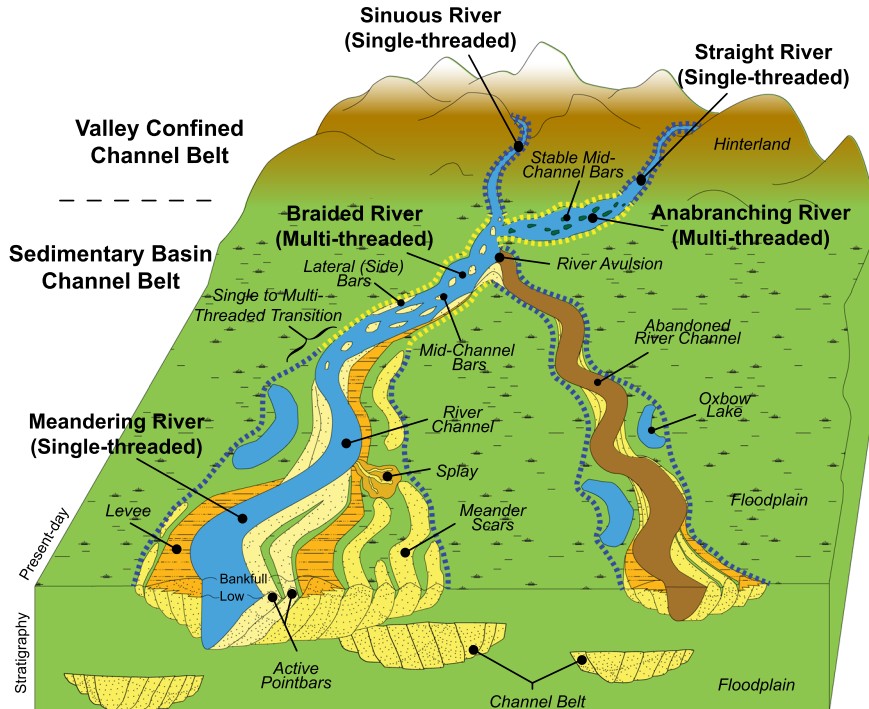

**Fig. 1 | Channel belt Terminology−schematic illustration of the channel belt extent that include the encompassing area of straight, sinuous, meandering, braided and anabranching river channels (active and abandoned) and its associated levees, bars, and overbank landforms.** The current Global Channel Belt model defines the observable extent of channel belts based on 30 m Landsat satellite imagery and its associated multi- or single-threaded river channel character as depicted by the dotted yellow and blue lines, respectively. This method can identify landform patterns that define the channel belt extent and transition between single and multi-threaded river types not feasible by traditional pixel-based classification techniques.

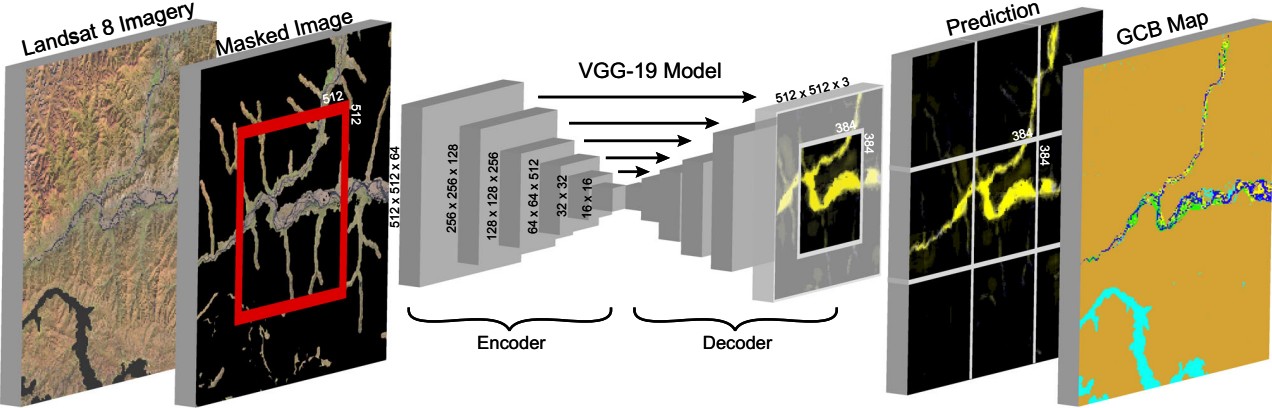

**Fig. 2 | Global Channel Belt model−The Global Channel Belt (GCB) model is based on a VGG-19[36] machine learning algorithm for pattern recognition of channel belt extent and single- or multi-threaded river character.** The algorithm uses 512 × 512 pixel tiles of Landsat 8 images masked for non-riverine regions using a series of convolutions and upscaling functions to simplify the prediction for a 3-class prediction of the channel belt extent with an associated single-threaded river channel, multi-threaded river channel or a background value. The resulting channel belt prediction is used to describe the distribution of fluvial and lacustrine environments. See methods for more detail. Landsat-8 images courtesy of the U.S. Geological Survey.

in river widths over the past 36 years based on water surface change[21]. However, these studies do not map the extent of the channel belt which is crucial for understanding the areal extent of river support for different ecosystems and its impact on biogeochemical cycles, flooding, and water resource management. A major challenge in mapping the channel belt with traditional pixel-based classification techniques is in capturing the number of different planform features across a range of different climates, vegetation types and lithologies.

Here we build the global channel belt (GCB) map to characterize the extent of channel belts for a cloud- and snow-free Landsat 8 composite image for the year 2020 consisting of 151,723 image scenes.

By implementing pattern recognition (Fig. 2) trained to 370 manually interpreted river systems across a range of different climates and geographical regions, we can predict the extent of channel belts to a 94% accuracy (see Methods; Supplementary Figs. 1 and 2). The machine learning algorithm will also predict the single- or multi-threaded character of the associated river channel on a spectrum ranging between a 0 to 100% confidence. The confidence shows the likelihood of a river channel being single- or multi-threaded, and where 0% implies neither river type. Furthermore, we define sub-environments within the channel belt extent including the active river channels and oxbow lakes in 2020 and the river channel

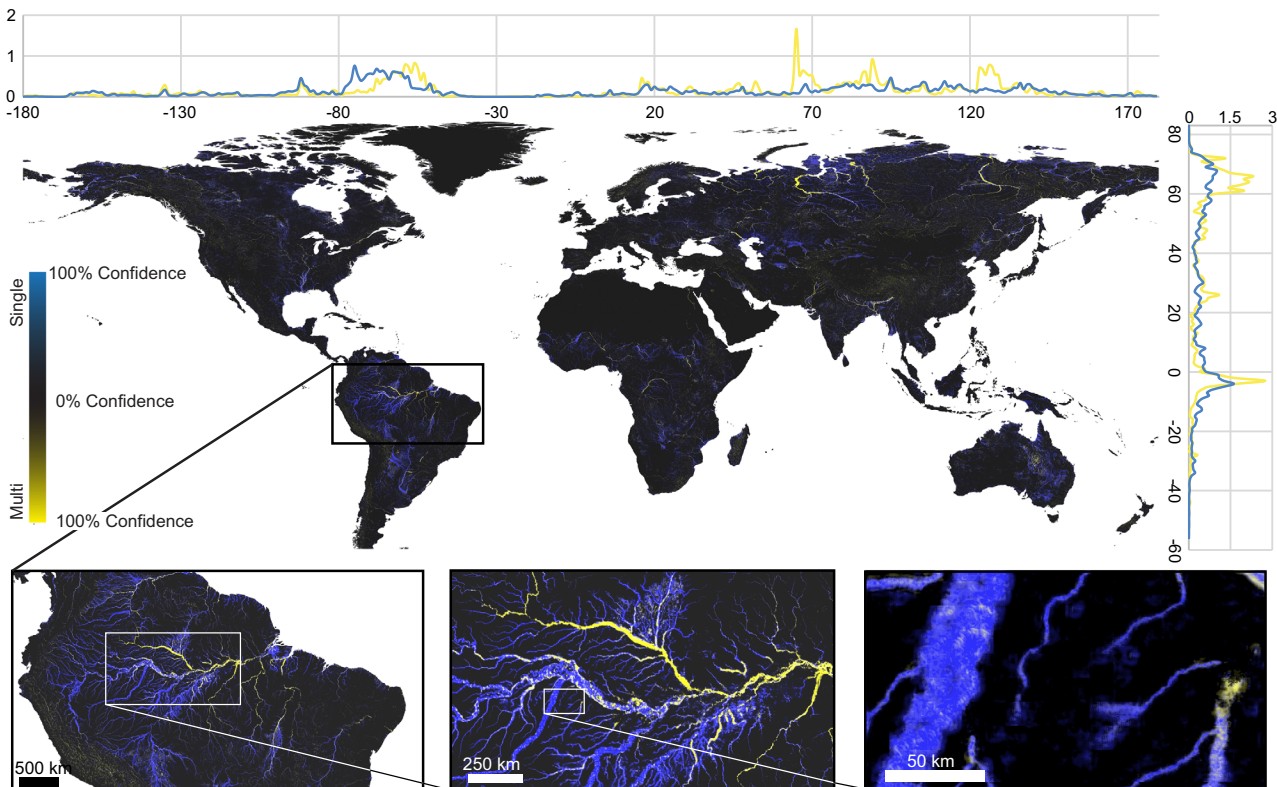

**Fig. 3 | Global scale analysis of channel belt extent**–map shows the predicted extent of channel belts and the single- versus multi- thread planform character of its river channel. Latitudinal and longitudinal plots show the proportion of single versus multi- threaded river channels as a percentage of the total. See data availability section for a detailed interactive map and publicly available dataset.

migration over the past 36 years (1984-2020). To increase the number of training images, we implement traditional data augmentation techniques[22] combined with two years of image acquisitions (2016 and 2020). In addition, the number of training images required to confidently identify planform features of the channel belt from non-riverine landforms is reduced by implementing a targeted image classification approach by masking the original Landsat 8 imagery for non-riverine regions (Fig. 2; see Methods). The GCB model provide new datasets for ecosystem accounting, freshwater resource management, analysis of biogeochemical cycles and flood mitigation studies, and the methods developed show the potential of a machine learning approach in classifying landforms on a global scale.

## Results
### Global Channel Belt map
The observable extent of channel belts covers a surface area of $30.5 \times 10^5$ km² (Fig. 3; see data availability section for interactive map), nearly 7 times larger than the documented extent of rivers[3]. This value is based on the reported 50% confidence interval of the GCB model at an ~1 km resolution (see Methods for validation). Globally, 37% of channel belts are in Asia ($11.4 \times 10^5$ km²), followed by 23% in South America ($7.4 \times 10^5$ km²), 14% in North America ($4.3 \times 10^5$ km²), 12% in Africa ($3.6 \times 10^5$ km²), 7% in Europe ($2.0 \times 10^5$ km²) and another 6% in Oceania ($1.8 \times 10^5$ km²; Supplementary Table 1). Multi-threaded river channels associated with the channel belt tend to dominate the larger rivers found in high latitude areas such as Siberia and northwestern Canada and Alaska, as well as tributary rivers in equatorial and temperate regions of the Amazon, Congo, Bangladesh, India, and Pakistan. Single-threaded river channels are more common across Africa, North America, South America, and Oceania with a lower occurrence in Europe and Asia (Supplementary Table 1 and Supplementary Fig. 3).

### Riverine and lacustrine environments
Based on the known extent of the predicted channel belt, we are further able to produce a new global classification of riverine and lacustrine / wetland environments (Fig. 4). Here we classify planform characteristics of waterbodies within the channel belt extent at the 30 m Landsat resolution. The map defines; (1) active river channels, (2) oxbow lakes, (3) extent of river migration from 1984 to 2020, and (4) the remaining channel belt extent including levees, bars, splays, overbank and abandoned river channel landforms.

The active river channels are defined by the pixels that create long elongated waterbody features (>~4.5 km long) based on an average annual water discharge level for 2020. Smaller disconnected waterbodies at the 30 m resolution within the channel belt are defined as oxbow lakes or smaller river reaches. River migration is defined by the maximum seasonal waterbody extent (>1 month of water detection) of the active river channel (including avulsions) based on a 36-year Landsat imagery time-series from 1984 to 2020[14]. The remaining area of the channel belt without an identified waterbody in 2020 or recent river migration since 1984 are defined as the remaining channel belt environment. Finally, waterbodies in 2020 that are defined outside the channel belt are classified as either lakes or wetlands (see Methods for further detail and validation).

Globally, the results show that the observable extent of channel belts covers a surface area of $30.5 \times 10^5$ km², which is similar to the extent of lakes and wetlands at $30.6 \times 10^5$ km². The distribution of lakes and wetlands are most prevalent in high latitude regions, whereas the channel belt environment becomes dominant in mid and low latitude regions (Fig. 4). Within channel belts, the active river channel extent in 2020 covered an area of $4.72 \times 10^5$ km² or 15% of the total channel belt area. Oxbow lakes or smaller rivers represent an additional $1.46 \times 10^5$ km² (5%), and an additional $2.70 \times 10^5$ km² (9%) represent the extent of river migration over the past 36-years of

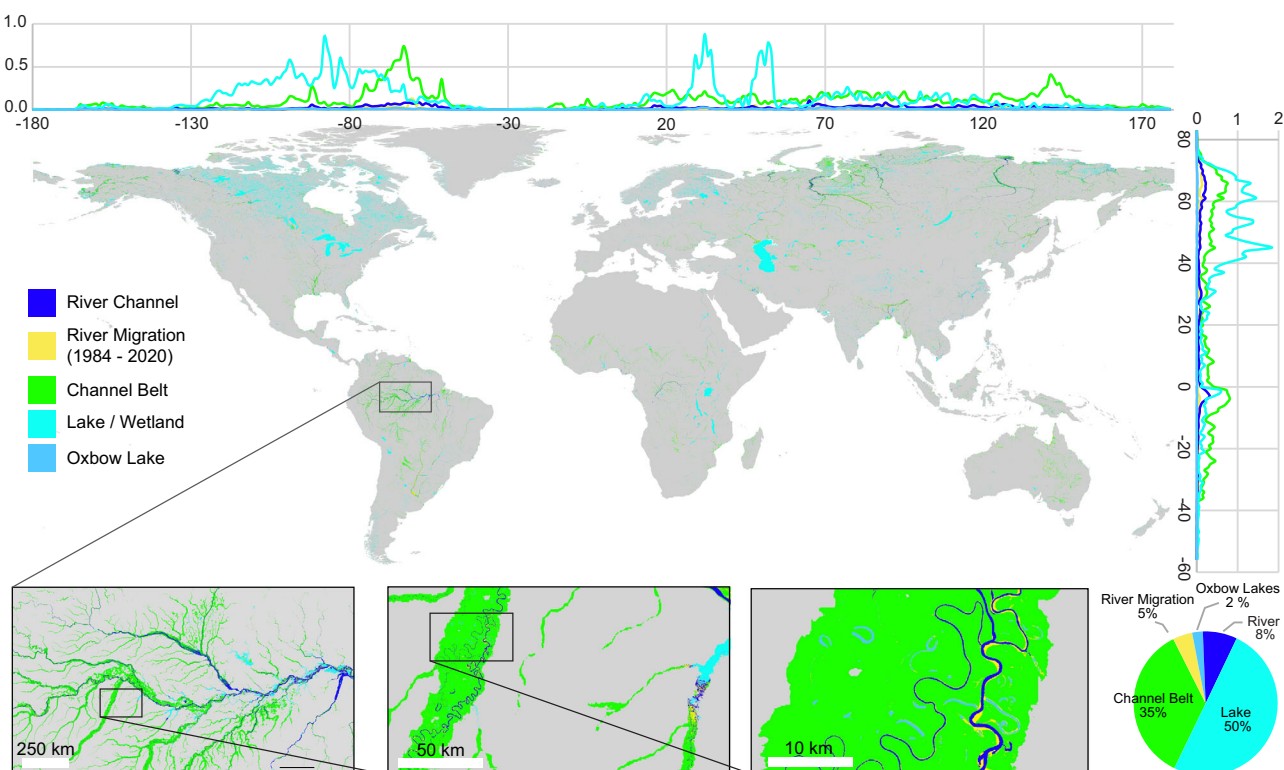

**Fig. 4 | Global scale analysis of riverine and lacustrine environments—map showing the distribution of rivers, lakes/wetlands, smaller rivers or lakes, channel belt and the active channel belt.** Latitudinal and longitudinal plots show the proportion of riverine and lacustrine environments as a percentage of the total. See data availability section for a detailed interactive map and publicly available dataset.

Landsat observations. The remaining $22.7 \times 10^5$ km² (71%) of the area represent the remaining channel belt environment without waterbody features including abandoned channels and associated overbank, levees and splays. The most actively migrating river systems over the past 36-years are associated with meandering rivers of equatorial regions as well as the larger braided river systems of the northern latitude regions and in foreland basins such as the Himalayas, Andes and Amur (Fig. 4). Based on the 50% confidence threshold, 52% of active river channels in 2020 were multi-threaded with the remaining 48% showing a more single-threaded character (Fig. 3, Supplementary Table 1).

### River channel characteristics

The hydrological, physio-climatic and tectonic conditions of the river channel by surface area are summarized in Fig. 5 (see Methods). Rivers with medium or lower long-term averaged water discharge rivers (<1000 m³ s⁻¹) represent approximately two-thirds of river channels by surface area. Another third of river channels are characterized by a high or very high (>1000 m³ s⁻¹) water discharge. In terms of morphology, very low, as well as high and very high-water discharge rivers are commonly multi-threaded at 56, 58 and 75%, respectively. In contrast, the global surface area of river channels with a low and medium water discharged are only 40% multi-threaded (Fig. 5a).

Nearly 30% of river channels are associated with very hot (>20 °C long-term averaged minimum air temperature of the coldest month) and high moisture (>0.125 climate moisture index, CMI) physioclimatic conditions in equatorial regions. River channels in these regions are slightly more single-threaded (56%; Fig. 5b). Cold (<−20 °C), low and medium moisture (<−0.4–0.125 CMI) regions also contribute a significant 20% of the total surface area of river channels and are 55% multi-threaded in contrast to equatorial climates. Multi-threaded rivers are dominant in warm and hot (−20–20 °C) low (<−0.4 CMI) moisture regions, warm (−20–5 °C) and medium (−0.4–0.125 CMI)

moisture regions as well as cold and warm (<−20–5 °C) high elevation (>750 m) regions at 60%, 60% and 65%, respectively.

When viewed by tectonic settings, 50% of river channels occur in passive margins, followed by foreland (27%) and intracratonic settings (16%) (Fig. 5c). Extensional/strike-slip and forearc settings combined define the remaining 7% of river channels. This distribution of river channels is similar to the distribution of the tectonic regions globally[23]. By morphology, we see that the proportion of multi-threaded to single-threaded river channels is relatively equal throughout the different tectonic regimes with intracratonic settings the most single-threaded at 56% and passive margins the most multi-threaded at 56%.

### Discussion

Global river analyses over the past few decades have primarily been based on hydrological river networks that delineate rivers as lines of river drainage based on digital elevation models[2,17,24]. More recently, with increased computational power, have global scale analyses[13] enabled the quantification of the surface area of river channels[3]. Yet the previous studies do not map the areal extent of channel belts that the current study shows are a significant region of riverine environments covering an estimated area that is 7 times larger than active river channels (Figs. 2 and 3). Similarly, no existing global land cover, land use or water surface area change map[14–16,25] captures the landforms that define the broader channel belt (Supplementary Fig. 4). As a result, existing interpretations of the channel belt have previously been based on manual interpretations[19,26] limiting our understanding of the larger riverine systems and their impact on biodiversity, climate and human livelihoods on a global perspective.

The new GCB model uses recent advances in pattern recognition to provide an objective and quantitative approach to identify landforms that define river channel belts. Machine learning is ideal to identify planform characteristics of river systems given that fluvial sedimentary processes leave distinct planform patterns such as

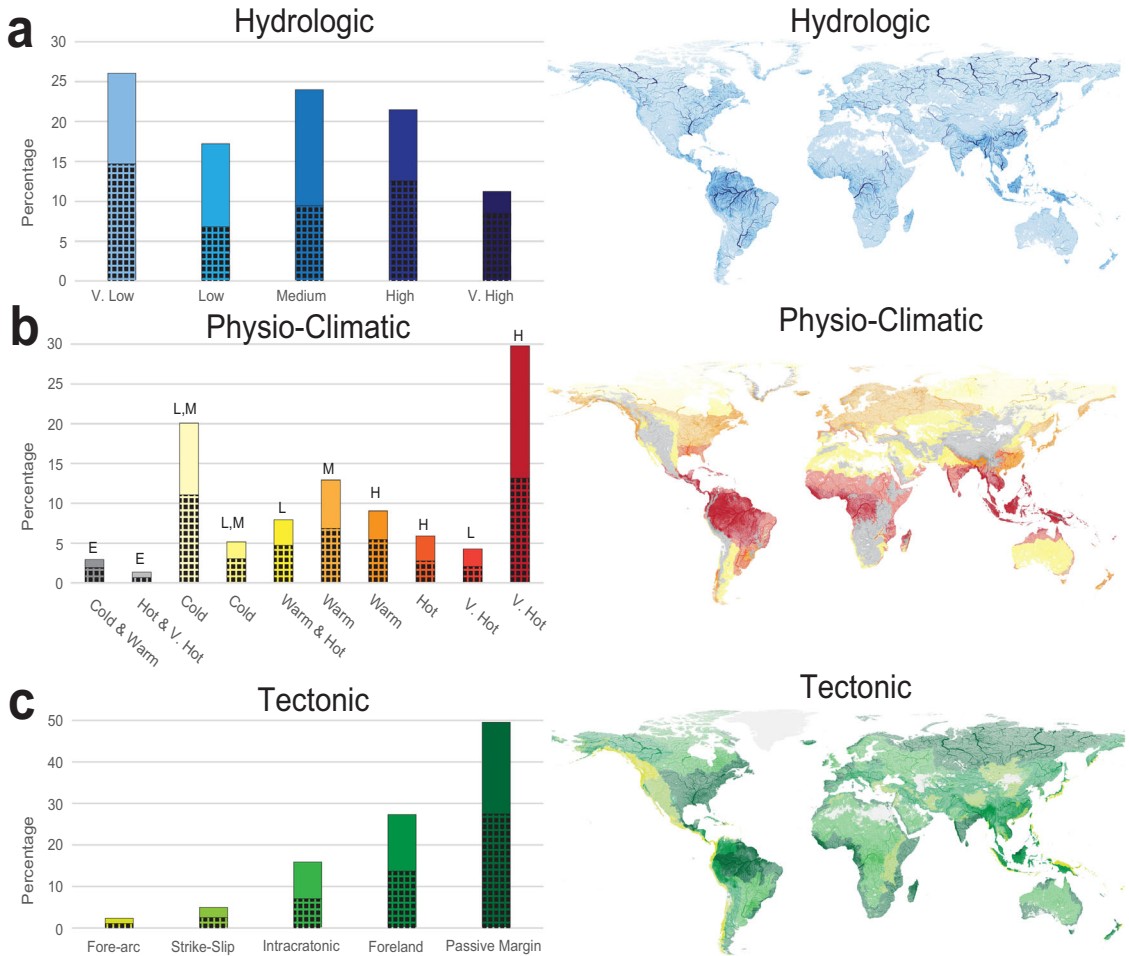

**Fig. 5 | Global river morphology distribution and controls.** The hydrological (**a**), physio-climatic (**b**) and tectonic (**c**) distribution and controls on rivers as a percentage of river surface area visualized based on river reaches[2]. The shaded region of each column shows the proportion of the total surface area of river channels defined as multi-threaded. Long-term averaged monthly river discharge is defined by Ouellet Dallaire et al.[2], as very low (0.1–10 m³ s⁻¹), low (10–100 m³ s⁻¹), medium (100–1000 m³ s⁻¹), high (1000–10,000 m³ s⁻¹) and very high (>10,000 m3 s-1). Physio-climatic conditions are defined by Ouellet Dallaire et al.[2],

based on three variables of temperature, climate moisture index (CMI; precipitation/potential evapotranspiration) and elevation. Categories show cold (<−20 °C), warm (−20–5 °C), hot (5–20 °C) or very hot (>20 °C) long-term averaged minimum air temperature of the coldest month and a low (L; <−0.4 CMI), medium (M; −0.4–0.125 CMI), high (H; >0.125 CMI) moisture index or a high (E; >750 m) elevation. Tectonic regimes are defined by Nyberg et al.[23], and include fore-arc, strike-slip (including extensional), intracratonic, foreland and passive margin settings.

meander scars and lateral accretion surfaces that can be trained and learnt by convolutional neural networks. Previous algorithms used to study surface area of rivers[3] are based on the distinct spectral characteristics of water alone based on methods primarily developed two decades prior[27]. Our approach to increase the number of training images through data augmentation and multi-year image acquisitions combined with a targeted masked image classification technique (Fig. 2) has allowed for a small training database of 370 river systems to suitably train the machine learning algorithm (see Methods). The resulting accuracy versus time-efficiency for analyzing remotely sensed imagery at a global scale shows the potential to apply similar methods to map a range of different sedimentary landforms that can be used to relate landforms to causative surface processes.

As with any model, the accuracy of the prediction depends on the reliability of the data inputs. Inherently, the model is limited by the 30 m Landsat imagery resolution split into roughly 512 × 512 pixel (~15 km²) tiles needed for the machine learning computations at a global scale (Fig. 2; see Methods). A consequence is that the model may fail to recognize either the small- (<150 m) or large-scale (>15 km) landforms that define the channel belt extent. This may lead to under- or over-estimations at the

boundaries of the tiled images. Despite these limitations, our results suggest the GCB model compares well to previously described geomorphology of river systems at a roughly 1 km scale resolution (see Methods).

The resulting GCB model offers valuable new datasets to explore the impact of the river channel and its channel belt extent on flooding, ecosystems, climate, and water resource management. For instance, the landforms within the channel belt provides information on the evolution of a river system and its past flood events that can be used to predict future flood risks and potential flood extents[1]. The dataset also builds on previous efforts to create a baseline study of current riverine state for ecosystem accounting to measure the impact of climate change on the environment[2]. The extent of the river and its channel belt is also an important measure on the spatial impact of riverine environments on ecosystems which is not achieved by hydrological models[2,24] alone. Furthermore, by combining the river channel with hydrological and climatological observations (Fig. 5), we can relate the controls and behavior of river systems supporting different ecosystems.

Application of the GCB riverine and lacustrine environments (Fig. 4) may improve biogeochemical flux calculations by improving

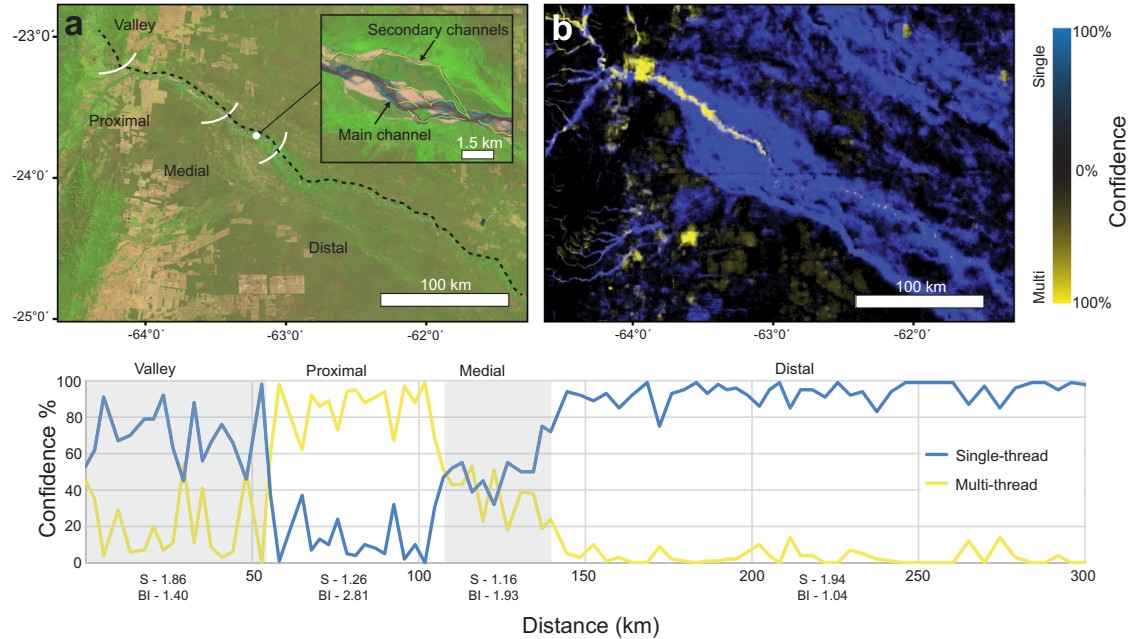

**Fig. 6 | Example river morphology profile. a** Shows the original 2020 Landsat 8 imagery of the Rio Bermejo in Argentina. The dotted line is the profile used to sample the morphology of the active river channel every 5 km's for the plot shown below. **b** Shows the resulting prediction from the Global Channel Belt (GCB) model indicating the gradual multi-threaded to single-threaded trend common in many distributive fluvial systems[18,19]. S refers to the sinuosity of the river channel (main channel length divided by the shortest valley path). BI is the braid index defined as the total sum length of channels divided by the main channel length. Landsat-8 images courtesy of the U.S. Geological Survey.

estimates of water surface area contribution to $CO_2$ outgassing and carbon capture by photosynthesis across the channel belt[3,7,28]. Moreover, the role of the river channel and its channel belt in carbon burial and release through sedimentation and erosion are known to be an important contributor to terrestrial greenhouse emissions, but relatively, poorly understood and quantified[7]. For instance, Repasch et al.[29] note variations in erosion of the channel belt of the Rio Bermejo, Argentina (Fig. 6) is an essential factor in carbon release of that river system. The GCB model and landform pattern recognition method provides delineations of riverine landforms important in terrestrial carbon flux cycles that may further help to constrain global biogeochemical estimates of carbon from rivers.

This study also presents the global-based analysis of the single-versus multi-threaded character of river channels by surface area at 48 vs 52%, respectively (Fig. 3). The dataset accurately show regional trends such as the aforementioned multi- (e.g., braided) to single- (e.g., meandering) threaded river channel character observed along the Rio Bermejo in Argentina (Fig. 6)[18,19,29]. Dimensions of channel belt landforms provide insights into the likely preservation and geometry of sedimentary deposits improving subsurface reservoir characterization in a range of applications, for example $CO_2$ sequestration and groundwater resource management[8,30]. Furthermore, the confidence-based predictions of the machine learning algorithm uniquely provide a quantitative estimate across a spectrum between end-member classifications (0–100%) allowing a more accurate portrayal of the variations that exist in river systems[11]. Although the results do not replace more detailed river channel metrics such as a braiding index, entropic braiding index[31] or sinuosity measurements[11], pixel-based classifications provide an area dimension and are less computationally expensive than vector based centerline and transect measurements[21,32]. The GCB model achieving global scale predictions classifying 2-degree regions in <1 h (see Methods), show the potential for near real-time pattern recognition which is needed to classify the dynamic nature of river landforms.

## Methods

### Training data mapping

A cloud and snow-free 2020 composite Landsat 8 imagery consisting of 151,723 scenes was created in the Google Earth Engine[13]. We manually interpret the Landsat imagery by overlaying each image with polygon interpretations that show the extent of the river and its channel belt at a 1:100,000 to 1:500,000 scale (Fig. 7; Supplementary Fig. 2). The interpreted area used in the training data range in size between 15 km² and 80 km² with channel belt widths that range between ~500 m to 30 km (e.g., Ob River).

Manual interpretation of the channel belt training images is based on the encompassing observations of; (1) active channels and associated bars, (2) overbank features such as levees and lateral splays, and (3) abandoned channels and its associated bar and overbank features (Figs. 1 and 7). The mapped planform features thus include the active river channel, point bars, mid-channel bars, lateral (side) bars, meander scars (e.g., relic point bars), levees, splays and abandoned channels (e.g., oxbow lakes, sediment filled channels)[10]. The extent of the channel belt will also include floodplain material deposited between the observed landforms associated with the river channel migration. Each interpreted polygon of the channel belt extent was given an attribute indicating the single- or multi-threaded character of the active river channel. Interpreted polygons were then converted to an image mask at the same 30 m Landsat image resolution with any non-interpreted region defined as a background value (Supplementary Fig. 2).

Single-threaded rivers are defined based on observations of either a straight, sinuous or meandering river channel morphology and its associated bar and overbank features. Figure 7a shows a typical example from the Rio Madre De Dios in Bolivia of an active meandering river channel and associated point-bar development with oxbow lakes and meandering scar landforms that define the channel belt extent. The Rakaia River in New Zealand and the Yukon River in Alaska are examples of multi-threaded rivers defined by multiple river channels creating well defined mid-channel bars and

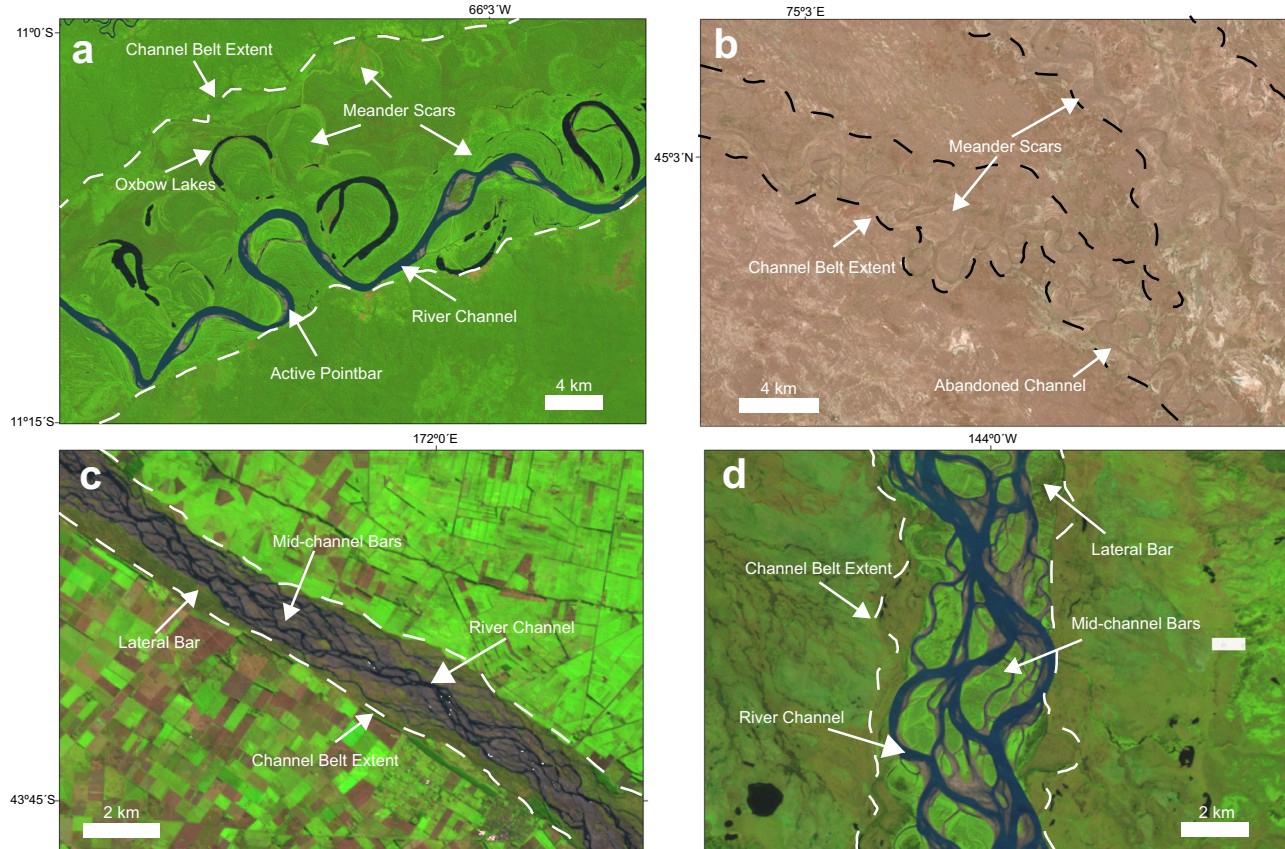

**Fig. 7 | Planform character of channel belts—example landforms that define the extent of the observable channel belt training images from 2020 Landsat 8 imagery. a** Single-threaded Rio Madre De Dios in Bolivia, **b** abandoned single-threaded Ili River in Kazakhstan, **c** multi-threaded Rakaia River in New Zealand, **d** multi-threaded Yukon River in Alaska, USA. Refer to main text for further detail. Landsat-8 images courtesy of the U.S. Geological Survey.

lateral side bars (Figs. 7c, d). In the case of a channel belt without an active river channel, we utilize the landform features to identify the likely river character (e.g., meander scars and abandoned meandering river channel) or refer to the adjacent active river channel for reference as was the case for the Ili River in Kazakhstan (Fig. 7b). The single- versus multi-threaded river classification is less reliable for an abandoned river channel (Fig. 3). Hence the reported values of single- versus multi-threaded rivers is based solely on the surface area of the active river channel (Fig. 4), although we include the full predictions in Supplementary Table 1 of the Supplementary Material for reference.

**Sampling selection and data augmentation**

A total of 790 localities were selected for training, 370 are riverine examples and an additional 420 localities are non-riverine regions covering a range of different climates, vegetation and landcovers. The location of the training images are randomly selected on the Earth's surface using several iterations to optimize the accuracy versus computational needs to train the algorithm. In total, we collected a database containing 1090 images at a 512 × 512 tiled resolution of both riverine and non-riverine examples for training. The Landsat 8 composite image averaging the pixel values gathered over the year likely represents a mean annual water discharge. While this is a source of uncertainty, a lack of data on months of low water discharge in global river systems prevents a more targeted image selection approach as noted in previous studies[3]. However, this issue is mitigated based on the pattern recognition approach classifying not only the river, but also the planform character of its channel belt as confirmed by the validation results (see Validation, Accuracy and Comparison section).

To further increase the number of images, we extract both 2016 and 2020 Landsat-8 imagery to increase the total number of scenes to 308,253 scenes and thus also increase the training dataset to 2180 images. The assumption is that both the spectral signatures and river morphology will be different for each year for the machine to learn. We limit this approach to 2 years to prevent overfitting the model and to reduce the computational requirements for the machine learning algorithm. In addition, we apply a series of common data augmentation techniques[22] to the images by randomly cropping between 70 and 100% of the original image, rotation between 90 and 360 degrees, and randomly flipping the resulting image. Given the scale invariance of river systems, the subsequent cropping, rotation and flipping augmentations will respect the morphology of the landforms that define the channel belt while helping the model predict at different scales.

**Global scale landsat-8 imagery processing**

To reduce the number of images required to confidently identify river morphologies from non-riverine features, we implement a targeted image classification approach by masking the original Landsat 8 imagery for non-riverine regions (Fig. 2). We remove Landsat imagery pixels that contain a mean slope >2 degrees within a 270 m window (or ~3 pixels) based on the 90 m resolution MERIT Digital Elevation Model[33]. Mountainous rivers in confined valleys were included by adding a 300 m radius (or ~10x Landsat imagery resolution) around river network lines with an upstream area >50 km² and a water discharge >0.1m³/s based on the free flowing rivers dataset[17].

In addition, oceans identified by the Global Shoreline Vector[34] and previously defined lakes[35] >10 km² were masked. This ensured that large waterbodies greater than the 512 × 512 (~15 km²) tiled resolution used in the machine learning predictions were correctly identified

(Fig. 2). Combined, these steps significantly reduced the number of training images required to identify the bounds of river channel belts in both mountainous and lowland regions. Finally, we only consider a false color RGB image using bands 6,5 and 4 of the Landsat 8 imagery to reduce the number of required input parameters and to be suitable for pre-trained machine learning models.

## Machine learning model

The machine learning model was built in Tensorflow/Keras from a pretrained VGG-19 model[36] on the ImageNet dataset[37] with a custom decoder involving a series of 5 upscaling, convolutions and ReLU activation functions (Fig. 2). The model was run with a batch size of 32 for 28 epochs based on a 3 run early stopping procedure on the reported validation accuracy. The 2180 images in our dataset are split into a training dataset for learning and validation dataset to test using a 70:30 ratio, respectively. Each epoch is refined based on an Adam optimizer and loss measured by a sparse categorical cross entropy. The model was trained on the Google Cloud AI platform using a n1 high-memory machine containing 64 virtual CPUs and 416GB of memory. The resulting model contains 21,353,943 parameters representing the internal variables of the machine learning algorithm (e.g., convolutions) used to objectively classify the Landsat imagery (Fig. 2). Each parameter is created and assessed by the machine learning algorithm itself to design the best model based on the available training and validation dataset.

To apply the model, the algorithm requires a $512 \times 512$ image input and creates a $512 \times 512$ image prediction containing 3 layers of probability (0–100%), one for each of three categories: (i) single-threaded, (ii) multi-threaded and (iii) background category. To limit potential edge effects in the resulting prediction, we export the Landsat imagery for Tensorflow as a series of $512 \times 512$ tiles with a 128 pixel overlap to keep the central $384 \times 384$ pixels for the resulting output (Fig. 2). To process the vast amount of data, we further split the data into 5064 2-degree tiles ($\sim$222 km²), each with its own 0.1 degree overlap and run the model on five virtual machines on the Google Cloud Platform. Tile based processing across multiple virtual machines allow for the high-resolution global scale predictions at the 30 m resolution to be processed in $\sim$32 h or <1 h per 2-degree tile. By combining the tiled predictions, we can produce a seamless map of global channel belt extents (see data availability section).

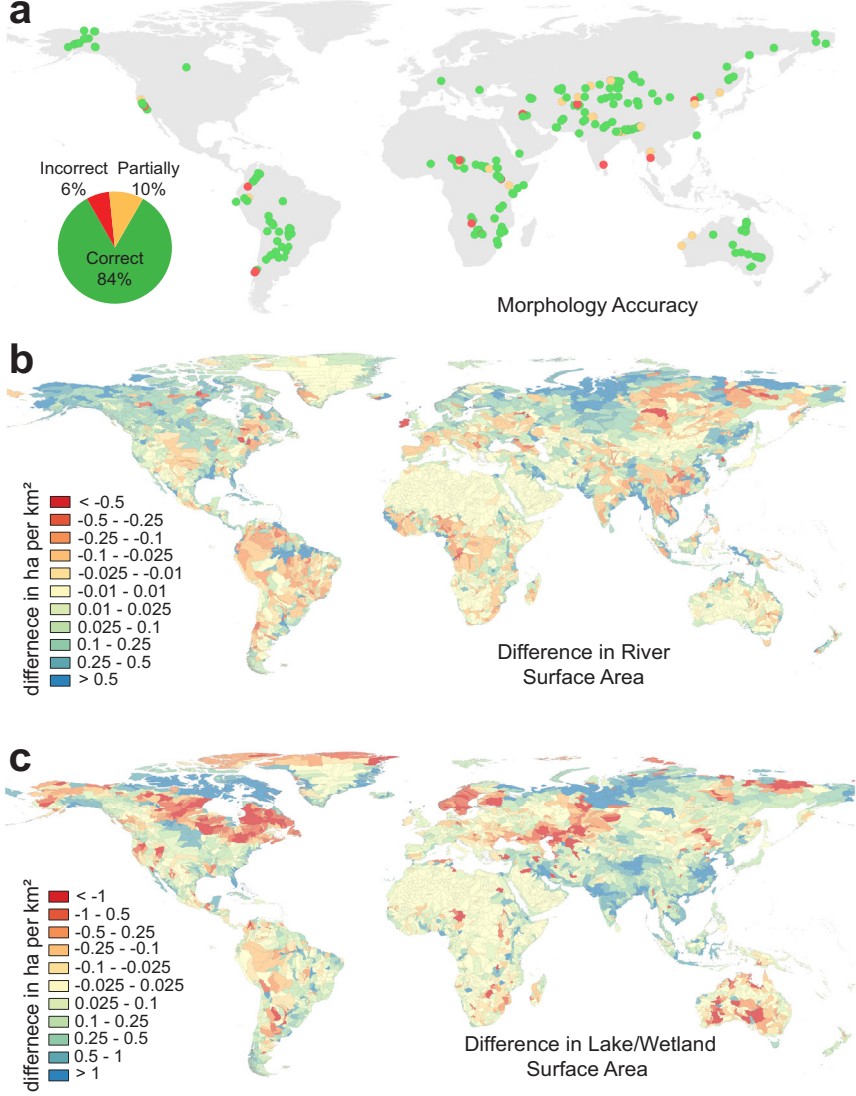

**Fig. 8 | Example river morphology profile. a** Accuracy of the Global Channel Belt (GCB) morphology compared to manual classifications by Hartley et al.[18], (**b**) difference in river surface area per km² within each sub-catchment between the GCB and the Global River Widths from Landsat (GRWL) dataset[3] and (**c**) difference in wetland/lacustrine surface area per km² within each sub-catchment between the GCB and the HydroLAKES datasets[35].

## Riverine and lacustrine sedimentary map

The riverine and lacustrine map defines the active river channel in 2020, 36-years of river migration from 1984 to 2020, smaller rivers or oxbow lakes in 2020 within the channel belt and the extent of lakes/wetlands in 2020. The extent of the entire channel belt is classified based on the 50% confidence boundary of the GCB model. The extent of river migration is defined by at least 2 years of seasonal (>1 month) water occurrence defined by time-series analysis of 36-years of Landsat imagery provided by the Global Surface Water Map v1.3[14]. For the average waterbody extent of 2020, we use the modified normalized difference water index (MNDWI) in Eq. 1 with a 0.6 index threshold on a 2020 Landsat 8 composite following the same established procedure as many previous studies[3,14,27].

$$MNDWI = green - SWIR/green + SWIR \qquad (1)$$

where green is the green band and SWIR is the shortwave infrared band. Large bodies of water >150 pixels connected within a $3 \times 3$ rectangular search window and at least a 10% channel belt confidence are assigned as the active river channel classification. Smaller bodies of water with an area <150 pixels within the channel belt are assigned as smaller river reaches or lakes. This class represents smaller river reaches that are typically disconnected at the 30 m Landsat resolution or smaller oxbow lakes that are a part of the channel belt environment. Finally, lakes and wetlands are defined as those permanent waterbodies with an area >4 pixels that lie outside the defined channel belt environment and within 100 m from the defined coastline[34]. This additional threshold was chosen to remove small clusters of pixel classifications that are difficult to identify as a waterbody based on Landsat imagery resolution.

## River channel characteristics

To define characteristics of the active river channel, we combine the GCB map in our study with existing data on hydrologic, physio-climatic[2] and tectonic[23] descriptions (Fig. 5). Given that the hydrological and physio-climatic descriptions of the GloRiC dataset[2] describe only the river reach, we expand that information to river extent by summarizing the information within sub-catchments of the HydroSHEDS level 12 product[24]. The maximum river discharge and largest sum of river reach length by climate within each sub-catchment are assigned a pixel classification that is subsequently related to the surface area of the GCB product. For the tectonic classification, the catchment delineations of the GTSC dataset[23] are overlain on the GCB map for analysis.

## Validation, accuracy and comparison

The machine learning classification of the channel belt extent shows a 96% accuracy to the training dataset and a 94% accuracy to the validation dataset with a loss of 0.13 and 0.15, respectively. Compared to the 415 manually described river morphologies by Hartley et al.[18], 170 were below the resolution of the GCB model and excluded from comparison. Of the remaining 245 examples, the single- versus multi-threaded prediction of the GCB model achieves an 84% accuracy (Fig. 8a). Another 10% of the locations were partially correct capturing one aspect of the river morphology while only 6% were incorrect. The channel belt width of the 170 excluded examples range between 10 m and 1300 m with a mean of 167 m (+/− 197 m) and a 95% confidence interval at 623 m. Hence, while the resolution of the Landsat imagery is defined at 30 m, several pixels are required to identify landforms that define the river and its channel belt, thus lowering the resulting resolution of the GCB model to -1 km width.

The current GCB model shows a river surface area of $4.72 \times 10^5$ km² (Supplementary Table 1) compared to the previously reported $4.7 \times 10^5$ km² of Allen and Pavelsky[3]. Spatially, the discrepancy in river surface area is shown to be higher in high latitude regions and lower in mountainous regions (Fig. 8b). This is likely since the river channel belt is less distinct in these regions and that the current study is based on an averaged river water discharge compared to a high-water discharge river surface area of the previous study. In total, the GCB model captures 91% of the river delineations by Allen and Pavelsky[3] within the extent of the channel belt predictions.

Compared to the previously reported extent of lakes[35], the current study shows roughly a 4% larger surface area at $30.6 \times 10^5$ km² (Supplementary Table 1). The most significant increase in lake surface area occurs along coastal wetlands and ephemeral salt lakes in South America, India and the Arctic that were not considered in the previous classification (Fig. 8c). In addition, lake levels have increased in the Himalayas due to reported increase in glacial melting[38]. A decrease is most prominent in central Asia and Australia associated with water loss over the past three decades due to climate change and excess water demand[14]. An underestimation of lake extent in the Canadian shield and Scandinavia is likely a result in the overestimation of the channel belt extent used to define lakes in the current study. Overall, the new riverine and lacustrine map show a good correlation at the global scale with <1% difference in pixels per km² (Fig. 8).

## Data availability

The Global Channel Belt (GCB) data generated in this study have been deposited in the Zenodo database under accession code https://doi.org/10.5281/zenodo.7680163. An interactive map is available at bjornburrnyberg.users.earthengine.app/view/gcbm.

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

## Acknowledgements

We would like to thank Adrian Hartley for kindly providing the locations of the manual interpretations of river morphologies used to compare to the current GCB model. Linling Chen is thanked for reviewing an earlier manuscript version. This study was funded by the Architectural Element Characterization of Fluvial Systems project by AkerBP ASA. Landsat-8 images courtesy of the U.S. Geological Survey.

## Author contributions

B.N., R.R. and R.L.G. conceived the original idea. B.N. designed the methodology and performed the data analysis. B.N., G.H., R.L.G., R.R. and J.A. interpreted the results. B.N., G.H., R.L.G. and R.R. wrote and edited the manuscript.

## Funding

## Competing interests

The authors declare no competing interests.
