## [Peer Review File · Nature Communications]

Global Scale Analysis on the Extent of River Channel BeltsReviewers' comments:

Reviewer #1 (Remarks to the Author):

This study uses machine learning and image processing to classify channel belts globally as meandering or braided. The study relates the classified channel belt information to other geographic information like population, tectonics, climate, and hydrologic factors. While the analysis was very impressive in terms of its magnitude and advanced methods, there were several significant problems with the study that I identified:

First, “channel belt morphology” was never defined. Does a “braided channel belt morphology” mean that the river was braided that deposited the sediments that created the channel belt? Does it mean that the planform of the channel belt is braided, not meandering? Or is there some other meaning that I didn’t catch? Since this was the main finding of the study, unfortunately it was hard for me to understand what the entire study was about or why it was important.

Second, related to this first serious flaw, the Methods do not adequately describe what was done in this study. As far as I could tell, machine learning and image processing was applied only to Landsat 8 imagery. Then this was used to define the extent of channel belts, which I assume is essentially a floodplain map (?), and then this channel belt map was classified into the braided and meandering types. Were the only predictor variables derived from Landsat 8 imagery? There were a lot of other datasets mentioned in the Methods, but evidently they were not used to train the algorithm. Additional questions I had included: How was the initial river mask created? What are the 21,353,943 parameters referred to in the main text?

Given the lack of clarity of the setup, methods and results, and the fact that the data are not publicly available (e.g. GRM could be loaded into Google Earth Engine for the reviewers to view the product), I cannot recommend that this paper be published in its current form. Please publish the data product for the sake of open science.

If the authors wish to continue working on the manuscript, I suggest including discussion of the following recent articles:

Tian et al. (2022) Quantitative relationships between river and channel-belt planform patterns
<https://doi.org/10.1130/G49935.1>

Feng et al. (2022) How Have Global River Widths Changed Over Time?
<https://doi.org/10.1029/2021WR031712>

I also added several line-edits and minor comments to the uploaded word document.

Reviewer #2 (Remarks to the Author):

The authors offer a global classification of river planforms into categories of braided and meandering, based on a remote sensing analysis. The main contribution here is methodological – it is quite impressive and astonishing that tools are now available to assess channel planforms on a global scale. The methods used by the authors are important and will serve to inspire additional studies of global river morphology, an important step in our ability to map and understand rivers across our planet.

I wish I could be equally impressed with the contribution to scientific knowledge represented by this manuscript. I found the geomorphic content of the results rather limited, and I remain unconvinced that the classification approach is particularly informative. Here are some of my concerns with the current version of the manuscript:

1. The simple classification of rivers into “braided” or “meandering” seems overly simplistic. In my experience, most rivers fall into neither category. A classification that would include additional categories would seem to be more realistic. For example, single-thread alluvial rivers can “sinuous” or “straight”, hence neither meandering nor braided. All of these categories can be anabranching or single-thread, a very important division of river types that I think the author’s methods should be able to detect. It may also be important to identify rivers that are confined by valley walls or terraces, or that are alluvial or non-alluvial (semi-alluvial and bedrock channels are commonly cited alternatives to alluvial rivers). Some of the issues with the current manuscript may simply arise because the terms “braided” and “meandering” are never clearly defined by the authors, but I think they could have adopted a classification scheme based on planform attributes that would be more useful from a scientific perspective.
2. What are meandering rivers, according to the authors? Generally there is a minimum sinuosity required for a river to be meandering, otherwise it is considered straight or sinuous. The authors should specifically cite how they define and identify meandering rivers.
3. What are braided rivers, according to the authors? I understand this term to refer to rivers with multiple thalwegs created primarily by well-defined alluvial bars, most of which are typically submerged when the river channel is filled with water (i.e., at bankfull stage). If the water level is high, I am not

confident that a braided river can be observed from a masked aerial image, since the bed of the stream (required to identify the multiple thalwegs of a braided stream) are underwater. How do the authors solve this problem? How many braids (usually summarized by a braiding index or other quantitative term) are required before the channel is identified as “braided”? This seems essential to clarify. Furthermore, how do the authors distinguish anabranching channels from braided channels? These are different categories of rivers according to nearly all contemporary geomorphic classification schemes, but may be difficult to discriminate clearly from aerial imagery.

4. What are the scale limitations of the analysis? Large rivers have different geomorphic properties from smaller rivers, and therefore it is essential to clearly define the sizes of the rivers included in this analysis. The only reference I could find that clearly addresses this refers to “known” rivers with a minimum drainage area of 50 sq. km and a minimum discharge of 0.1 cms, neither of which seem like realistic lower limits of resolution for a global analysis.

In summary, I believe that these issues should be addressed before this manuscript can be published.

Jim Pizzuto

Dept. of Earth Sciences

University of Delaware

USA

Some detailed comments, keyed to the text:

1. Title: Title should identify the aspect of the geomorphology that is addressed by the research. This is too broad.
2. Line 7: “their channel belt” should be “its channel belt”.
3. Line 7. “hydrological processes” seems too limited. Tectonics, time, humans, etc.
4. Lines 11-12. Please indicate the scale limitations of the analysis. This research cannot cover all channels at all scales across the globe.
5. Lines 13-14. The division into meandering and braided seems too limited. Anabranching? Sinuous? Other types of channels?
6. Line 15. I do not understand what a “braided” channel belt is. This term is applied to a river channel, not to its channel belt.
7. Lines 41-42. Please correct poor grammar.

8. Lines 43-44. This is confusing. The planform classification does not require identifying the channel belt.
9. Line 49. Selection criteria?
10. Lines 38-57. Temporal extent? Over what time periods are the images available?
11. Line 59. Are the study areas selected randomly? Does the algorithm look everywhere for rivers? Are these % values meaningful, or do they simply reflect the areas selected by the authors? Please clarify.
12. Line 83. 36 years....? But the temporal coverage of imagery has not yet been stated. Please clarify.
13. Line 115. But not more is stated about the channel belts...
14. Line 118. The relationship between “stability” and planform classification is very weak.
15. Lines 117 – 118. Other types of rivers besides meandering and braiding are important to consider. Bedrock rivers?
16. Line 130. Classifications are not "confident"; people are, or are not, confident. Please revise to improve word choice.
17. Line 134. “Or....”. This is not a complete sentence.
18. Line 157. Please give a numerical or more descriptive interpretations of “small” and “large”. These relative terms are not very helpful.
19. Line 194. “known” hydrological drainage patterns...? Please explain what these are and how they are used in the analysis.
20. Lines 195 and 196. Please explain these size limits. I am very skeptical that the analysis can include these very small rivers – the resulting data set, on a global scale, would include millions and millions of stream channels. Are these all really included in this analysis?
21. Figure 3. The mapped river channels do not show up well on the black background of the continents on my version of this figure. Perhaps this figure can be redesigned to better illustrate the extent of the channels identified by the author’s analyses. Perhaps this type of figure looks better on a live computer screen, as opposed to the pdf document I am reviewing.
22. Figure 4. See comment above. Very light green river channels do not show up well here on the light gray background.

Reviewer #1

Q1 - First, “channel belt morphology” was never defined. Does a “braided channel belt morphology” mean that the river was braided that deposited the sediments that created the channel belt? Does it mean that the planform of the channel belt is braided, not meandering? Or is there some other meaning that I didn’t catch? Since this was the main finding of the study, unfortunately it was hard for me to understand what the entire study was about or why it was important.

A1 – The definition of the channel belt and its morphology may not have been properly defined in the original manuscript. We have added another paragraph in the introduction to clearly state the definitions used in the current study to classify the meandering and braided channel belt morphology. Here we define the channel belt morphology by the planform character of the channel belt. Braided river systems will display multiple thalwegs with alluvial bars in the form of mid-channel bars and lateral accretions. Meandering river systems are defined by meandering river channels creating point bar accretions, overbank deposits and creation of oxbow lakes.

Q2 - The Methods do not adequately describe what was done in this study. As far as I could tell, machine learning and image processing was applied only to Landsat 8 imagery. Then this was used to define the extent of channel belts, which I assume is essentially a floodplain map (?), and then this channel belt map was classified into the braided and meandering types. Were the only predictor variables derived from Landsat 8 imagery? There were a lot of other datasets mentioned in the Methods, but evidently they were not used to train the algorithm. Additional questions I had included: How was the initial river mask created? What are the 21,353,943 parameters referred to in the main text?

A2 – The heading “Planetary Scale Landsat-8 Imagery Processing’ in the Methods section describes the preprocessing steps of the imagery prior to its application by the machine learning algorithm. Indeed, Landsat 8 imagery bands 6,5 and 4 were applied in the machine learning algorithm. While additional bands and datasets may improve the resulting accuracy, currently this would require a significantly larger model due to all the possible permutations, that would make it computationally unfeasible to apply at a global high-resolution 30 m pixel resolution. Furthermore, a 3-band image allow for the application of a pre-trained model to be used to significantly reduce the number of training images required to identify river systems. In this circumstance we have applied the VGG-19 machine learning model. The additional datasets mentioned in the methods were used to define the river mask rather than as an additional dataset to train the algorithm. These pre-processing steps were explained under this heading.

The number of parameters in the resulting machine learning model will depend on its architecture, which in this case is based on a full convolutional neural network design of the VGG-19 model that include a series of convolutions, downscaling and upscaling procedures as illustrated in Figure 2. This may not have been clear in the original manuscript and has now been improved. In addition, with the improved definition of the channel belt (see Answer A1), the methods section of the manuscript now clearly explains the method for training and predicting river system extent and morphology.

Q3 - Please publish the data product for the sake of open science.

A3 – The datasets are publicly available including an interactive map as stated in the “Data Availability” section. However, this may not have been clear in the main text of the original manuscript which has been revised manuscript in the updated version.

Q4 - I suggest including discussion of the following recent articles: Tian et al. (2022) Quantitative relationships between river and channel-belt planform patterns <https://doi.org/10.1130/G49935.1>
Feng et al. (2022) How Have Global River Widths Changed Over Time? <https://doi.org/10.1029/2021WR031712>

A4 – These recent articles are good suggestions and we have added those to both the introduction and discussion.

Reviewer #2

Q1 - The simple classification of rivers into “braided” or “meandering” seems overly simplistic. In my experience, most rivers fall into neither category. A classification that would include additional categories would seem to be more realistic. For example, single-thread alluvial rivers can “sinuous” or “straight”, hence neither meandering nor braided. All of these categories can be anabranching or single-thread, a very important division of river types that I think the author’s methods should be able to detect. It may also be important to identify rivers that are confined by valley walls or terraces, or that are alluvial or non-alluvial (semi-alluvial and bedrock channels are commonly cited alternatives to alluvial rivers). Some of the issues with the current manuscript may simply arise because the terms “braided” and “meandering” are never clearly defined by the authors, but I think they could have adopted a classification scheme based on planform attributes that would be more useful from a scientific perspective.

A1 – The terms braided and meandering are indeed based on planform attributes and has now been clarified in the main text (see Answer A1 for reviewer #1). Furthermore, one of the significant benefits of a machine learning approach is that the classification yields a probability between the categories therefore providing a range which is more realistic as the reviewer rightly mentions. This was not clear in the original manuscript, and we have placed more emphasis on this point in the revised version. While we recognize that straight and sinuous river systems do exist, we were required to limit the number of potential classes in order to gather enough representative training images for each category. However, as we have stated in the new expanded discussion, the GRM model provides a foundation for further classification of river morphologies (straight, sinuous, bedrock, anabranching).

Q2 - What are meandering rivers, according to the authors? Generally, there is a minimum sinuosity required for a river to be meandering, otherwise it is considered straight or sinuous. The authors should specifically cite how they define and identify meandering rivers.

A2 – This has now been clarified in the manuscript. Here meandering river systems are characterized based on their planform character displaying meandering river channels creating pointbars, overbank deposits and oxbow lakes on its channel belt (also see Answer A1 for reviewer #1). The machine learning algorithm itself does not specifically define a minimum sinuosity threshold needed to identify meandering rivers systems but internally chooses a set of parameters needed to best match the training dataset. This has the benefit that the algorithm is objective providing a probability estimate rather than requiring one subjective threshold cut-off. We have also added another Figure (Figure 6) to illustrate how the GRM model may relate to a typical braiding index or sinuosity measurement.

Q3 - What are braided rivers, according to the authors? I understand this term to refer to rivers with multiple thalwegs created primarily by well-defined alluvial bars, most of which are typically submerged when the river channel is filled with water (i.e., at bankfull stage). If the water level is high, I am not confident that a braided river can be observed from a masked aerial image, since the bed of the stream (required to identify the multiple thalwegs of a braided stream) are underwater. How do the authors solve this problem? How many braids (usually summarized by a braiding index or other quantitative term) are required before the channel is identified as “braided”? This seems essential to clarify. Furthermore, how do the authors distinguish anabranching channels from braided channels? These are different categories of rivers according to nearly all contemporary geomorphic classification schemes, but may be difficult to discriminate clearly from aerial imagery.

A3 – We define braided rivers by multiple thalwegs creating alluvial mid-channel bars and lateral accretions. The number of braids required to define a braided river is objectively defined by the machine learning algorithm itself (see Answer A2) and has now been clarified in the manuscript. Furthermore, we have added an additional figure to illustrate this point (Figure 6) and how it may relate to a braiding index or sinuosity measurement. Given that the image chosen to classify river systems is based on a composite of images collected throughout the year, the radiometric character of the image represents an average thus likely mean water flow rather than a high discharge that would submerge the alluvial bars. This is however an uncertainty that has now been mentioned in the methodology, but the validation results shows that this is not a major concern. Anabranching systems are not specifically classified in the current GRM model but rather as a probability if the river is characterized by a more braided or meandering character. This provides a foundation for additional datasets to be applied, such as water surface change maps, to further discriminate these river morphology types. This has now been clarified in both the introduction and discussion of the manuscript.

Q4 - What are the scale limitations of the analysis? Large rivers have different geomorphic properties from smaller rivers, and therefore it is essential to clearly define the sizes of the rivers included in this analysis. The only reference I could find that clearly addresses this refers to “known” rivers with a minimum drainage area of 50 sq. km and a minimum discharge of 0.1 cms, neither of which seem like realistic lower limits of resolution for a global analysis.

A4 – The scale limitations are now clearer in the manuscript. This will depend on the resolution of the 30 m Landsat imagery and by the 512x512 (~15km) pixel tiled images used for training. The training images contain a range of different river channel belt sizes that vary in width from approximately 500 m to > 30 km in width. Our validation results suggest that the lower limit of the channel belt resolution defined by the algorithm is approximately 1 km considering several 30 m pixels are typically required to identify a river shape. The minimum drainage area and discharge mentioned in the methodology is used to simply identify the river mask and does indeed include the 8.5million + river reaches based on the free-flowing river dataset of Grill et al., (2019).

REVIEWER COMMENTS

Reviewer #2 (Remarks to the Author):

This revision continues to report on a global analysis of channel planforms and channel belts. The methods and results are novel because the entire globe is included, and when published, should generate considerable interest and spur additional work in this area. Like the first version, however, I still believe that the manuscript needs improvement before it should be published. Here are my primary concerns (some more serious than others):

1. The title refers to channel belt DEPOSITS, but the methods and results presented by the authors define channel belt landforms: no information on the underlying deposits can be determined from their methods, at least as they are explained in the manuscript. Correcting this issue is rather easily accomplished, but it is still important.
2. The manuscript fails to present a clear definition of a channel belt that can be used to explain how they are mapped. The definition on lines 30-32 refers to the sedimentary deposits created by the river as it migrates laterally, but remote sensing methods cannot detect these, and so a different definition is needed to support mapping from aerial sources. In the caption to Figure 1, the authors indicate that a channel belt “represents all features associated with the river and its lateral migration.” This is perhaps a start, but the reader will need to know what features the authors consider “associated with the river” and how they are recognized. Furthermore, it seems to me that the authors identify channel belts from a single image, rather than from a time series of images, and therefore they cannot be sure where the river has migrated laterally through time, so they should explain to the reader how this portion of the definition is used to identify channel belts. Also, what if the river migrates through avulsion; is this included in the authors concept of “lateral migration”?
3. The manuscript also lacks a clear and nuanced discussion of the two-fold classification approach to river planforms used by the authors. They discuss in general terms the possibility of sinuous and anabranching river systems, but apparently do not identify these in any of their training images, and they are not represented at all in the classification data set. Because they don’t measure sinuosity, it seems to me that they cannot distinguish between these sinuous and meandering rivers, and I would expect some of the rivers in the world to be single threaded but not meandering at the scale of this analysis. Furthermore, their methods also cannot detect the difference between anabranching and braided rivers, so it looks to me as if they have combined the two categories.

The last point is the real issue to me in the manuscript. I think that they are mapping single and multi-thread channels. The single thread channels are all treated as meandering (regardless of sinuosity) and any river with multiple threads or channels is considered to be braided (regardless of whether the flow bifurcates around a floodplain island (anabranching) or a mid-channel bar (braided)). This approach remains valuable, and it is worth publishing because of the global scale of the analysis, but the authors

need to be clear about what they are mapping: I am not convinced that they are mapping braided and meandering river channels as they claim.

I recommend at least three revisions to the manuscript. First, the authors should very clearly explain how they identify a channel belt and its extent. Second, they should clearly present their methods and classification, to explain what it is they are mapping and how they identify it, and how the classification is similar to or differs from classifications commonly used by fluvial geomorphologists. Third, to facilitate the second point, they should include some examples of the training data set they have classified manually and systematically explain the decisions that were made, so we can understand clearly how they think rivers should be classified according to their approach. Extended Figure 2 is a good start, but should be expanded to show the range of phenomena covered by the analysis, and a careful explanation should be included. Just as an illustration, the training images in Extended Figure 2 are never identified as either braided or meandering, so how can the reader understand the author's approach?

Jim Pizzuto

Dept. of Earth Sciences

University of Delaware

USA

Some detailed comments, keyed to the text.

1. Line 1, Title. No deposits are discussed or analyzed in this manuscript. Title should be revised.
2. Line 7, abstract. Need a definition of the channel belt, even in the abstract. This concept is not clearly understood or defined in general.
3. Line 30. Again, this discussion is not sufficient. Nearly all geomorphologists today recognize anabranching rivers as a planform attribute. The authors absolutely need to address this from the outset.
4. Line 34. But what IS a channel belt? How can it be recognized and defined, either from remote sensing or other attributes? A clear definition of a meander belt is still needed, such that it can be mapped. Without this definition, the reader cannot understand how meander belts can be recognized and mapped. Figure 1 provides an illustration of some definition, but the reader actually needs the definition to be made explicit.
5. Line 35. Poor topic sentence. Paragraph is not about meandering rivers, but is a general discussion of river planform types.
6. Line 38. No, sinuous rivers need not be entrenched.

7. Line 42. This last sentence seems out of place.
8. Line 50. Poor grammar.
9. Line 65. It looks like they are mapping single-thread vs multithread channels, not meandering and braided based on the examples in extended Figure 1.
10. Line 70. Results should report the characteristics of the manually classified training data. This way the reader will understand how the training method is guided by manual classification decisions.
11. Line 84. But what is a meandering channel belt? This has still not been defined.
12. Line 94. I don't understand what it means to be "connected". Please clarify.
13. Line 95. ?? is this channel migration or just changes due to changes in water level? What is meant by "seasonal"? This is not a standard definition of "active" that would be used by geomorphologists. The definition needs to be presented at the outset and carefully explained and justified.
14. Line 97. A very narrow definition of "abandoned". Not very useful geomorphically...
15. Figure 1. What is the white area between channel belts?
16. Line 326. The "backswamp" away from the channel belt is also a "feature associated with the river" - this is a poor definition.
17. Line 327. Are historical channel positions used to determine this? Please clarify. 37 years is not an adequate time frame for determining the "recent" extent of channel migration – decades, centuries, or possibly even millennia are needed for this, which is why the channel belt may not be readily defined based on channel migration. Another definition is likely needed.
18. Figure 6. Images are too small to convey useful information. Very hard to see the channel in B.
19. Figure 6 of x axis label of lower panel: What is the percentage? Percentage of...??? Please define.
20. Line 391. The channels with multiple threads (A and B) look anastomosing to me, not braided.

Reviewer #3 (Remarks to the Author):

Summary

This manuscript used a machine learning method to classify river channel belts globally on a spectrum from meandering to braiding via satellite imagery data. These classifications of river channel belts are then used to correlate with geological predictors like tectonic settings and climate zones.

Impression

Overall, the manuscript is written fine, and figures are clearer. The topic addressed here: river channel belt/broader floodplain is an important research area for several fields in geoscience. I want to acknowledge that I received a revised version of the manuscript and I have read through the reviews from and responses to the two anonymous reviewers. While the data produced here is of significant value, the current manuscript does not contain much thorough analysis that yield novel scientific findings.

Comments

The manuscript is trying to address two very difficult problems in geomorphology at once: 1-classification of river planform morphology (meandering versus braided) and 2-classification of river channel-belt/floodplain complexes. Because it is challenging to address these two problems in a single manuscript, the study ended up just adapting previous qualitative river planform classifications and hand mapping floodplain features in labeling the training data. As a result, the manuscript lacks a convincing motivation. The previous two anonymous reviewers asked many questions regarding various definition of river planform and channel belt morphology supports my rationale. To address all our concerns, the manuscript should really have a focal point.

My suggestion is for the paper to highlight the usage of machine learning method to map channel belt and the associated floodplain features (Figure 4), instead of focusing on classifying river planform type. Mapping channel belt and floodplain feature is a perfect place to use machine learning because current method is mapping by hand, and we also have relatively limited knowledge of floodplain sedimentary processes. On the other hand, we have well established physical theory based on sediment transport and fluid flow to explain river planform patterns. The state of art classification is using entropic braiding index (Tejedor et al., 2022). This method counts channel branch based on the amount of water and sediment discharge it carry.

Minor Comments

116 I won't call it controls, since there is no quantitative nor qualitative mechanistic explanation of how these factors impact river morphology

139 Geometrical measurements are not subjective...it is known that average river channel geometry is set by balance between boundary shear stress of the flow and critical shear stress of the bed sediment, see Tejedor et al., 2022 for detail.

194-195 what is an expert opinion

Figure 3 It is a bit hard to interpret here because much of the channel belt does not include an active river. Instead, Figure 4 is much more useful and instructive for geoscientists.

Response to Reviewer Comments

Reviewer #2

Q1 - The title refers to channel belt DEPOSITS, but the methods and results presented by the authors define channel belt landforms: no information on the underlying deposits can be determined from their methods, at least as they are explained in the manuscript. Correcting this issue is rather easily accomplished, but it is still important.

A1 – This is a good point and we have corrected the issue by changing the title to “Global Scale Analysis on the Extent of River Channel Belts”

Q2 - The manuscript fails to present a clear definition of a channel belt that can be used to explain how they are mapped. The definition on lines 30-32 refers to the sedimentary deposits created by the river as it migrates laterally, but remote sensing methods cannot detect these, and so a different definition is needed to support mapping from aerial sources. In the caption to Figure 1, the authors indicate that a channel belt “represents all features associated with the river and its lateral migration.” This is perhaps a start, but the reader will need to know what features the authors consider “associated with the river” and how they are recognized. Furthermore, it seems to me that the authors identify channel belts from a single image, rather than from a time series of images, and therefore they cannot be sure where the river has migrated laterally through time, so they should explain to the reader how this portion of the definition is used to identify channel belts. Also, what if the river migrates through avulsion; is this included in the authors concept of “lateral migration”?

A2 – We have changed the terminology that we have used to indicate the landforms that identify the channel belt extent for the machine learning algorithm to learn. We define these planform features as the river itself and associated bars that are actively accreting and/or migrating; 2) the immediate overbank, levees, splays and floodplain and 3) abandoned channels and associated bars, levees, splays and floodplain landforms.

The prediction of the channel belt extent based on the machine learning algorithm is indeed based on one composite image for the year 2020. The extent of the channel belt is subsequently used to define the movement of water within the channel belt based on existing time-series of water movement from 1984 to 2020 based on the Global Surface Water model. We have clarified this point in the updated text. In addition, we have clarified the text “lateral migration” by changing it to “river migration” to include any lateral migration of the river as well as river avulsions.

Q3 - The manuscript also lacks a clear and nuanced discussion of the two-fold classification approach to river planforms used by the authors. They discuss in general terms the possibility of sinuous and anabranching river systems, but apparently do not identify these in any of their training images, and they are not represented at all in the classification data set. Because they don't measure sinuosity, it seems to me that they cannot distinguish between these sinuous and meandering rivers, and I would expect some of the rivers in the world to be single threaded but not meandering at the scale of this analysis. Furthermore, their methods also cannot detect the difference between anabranching and braided rivers, so it looks to me as if they have combined the two categories.

A3 – While the original training image collection was based on meandering and braided rivers, we agree that applying this classification to the global scale is an oversimplification of the complexity that exists in river systems. To address this issue we have changed the terminology to single-threaded

and multi-threaded as suggested by the reviewer. In addition, we have placed less emphasis on the river morphology classification in general and focused more on the novelty of the machine learning approach in classifying river landforms (see also Reviewer #3 Q2).

Reviewer #3

Q1 - The manuscript is trying to address two very difficult problems in geomorphology at once: 1-classification of river planform morphology (meandering versus braided) and 2-classification of river channel-belt/floodplain complexes. Because it is challenging to address these two problems in a single manuscript, the study ended up just adapting previous qualitative river planform classifications and hand mapping floodplain features in labeling the training data. As a result, the manuscript lacks a convincing motivation. The previous two anonymous reviewers asked many questions regarding various definition of river planform and channel belt morphology supports my rationale. To address all our concerns, the manuscript should really have a focal point.

A1 – Mapping the channel belt is indeed difficult and likely a main reason why it has not been mapped on a global scale previously. To address this, we have narrowed our focus to two main aspects in defining the channel belt extent and the advantage of the machine learning approach.

Q2 - My suggestion is for the paper to highlight the usage of machine learning method to map channel belt and the associated floodplain features (Figure 4), instead of focusing on classifying river planform type. Mapping channel belt and floodplain feature is a perfect place to use machine learning because current method is mapping by hand, and we also have relatively limited knowledge of floodplain sedimentary processes. On the other hand, we have well established physical theory based on sediment transport and fluid flow to explain river planform patterns. The state of art classification is using entropic braiding index (Tejedor et al., 2022). This method counts channel branch based on the amount of water and sediment discharge it carry.

A2 – This is an excellent suggestion and in the updated manuscript we have placed more emphasis on the machine learning method for classifying the channel belt extent. Our initial intent to also classify the river morphology was not meant to replace more detailed river morphology classification techniques but rather to compliment those existing tools. We have updated our manuscript accordingly to discuss the advantage of the machine learning approach in classifying both the channel belt extent and river characteristics (single- vs. multi-threaded).

REVIEWERS' COMMENTS

Reviewer #3 (Remarks to the Author):

Overall, the manuscript made substantial adjustments to refocus the research topic and to clarify various definitions of fluvial geomorphologies, such as channel belt extent and single versus multithread channels. In addition, the manuscript shares various maps generated during the research via an online repository. This open dataset, findings, and machine learning methods used in this work should be important contributions to the broader geoscience community. I have some very minor comments below. I very much appreciate the revision.

L12 percentage by area? Please be explicit.

L58-59 Can you provide an example of how to interpret this confidence? For example, what does 56% confidence in identifying the multi-thread channel river mean? What would this river look like? This is better explained in L114-115. So, the confidence here implies the likelihood that a river is multithread or single thread for a given land pixel. 0% confidence means it is neither type of river. Figure 3 made the concept easy to understand, but I suggest improving the writing here to make it clearer.

L81 try "as well as."

L126 cold (<-20 °C), missing a negative sign, same is L130. Please also check the others, like C.MI

L196, I would add "(e.g., braided)" and "(e.g., meandering)"

Reviewer #3

Q1 - L12 percentage by area? Please be explicit.

A1 – This does indeed refer to percentage by area and has now been clarified in the manuscript.

Q2 - L58-59 Can you provide an example of how to interpret this confidence? For example, what does 56% confidence in identifying the multi-thread channel river mean? What would this river look like? This is better explained in L114-115. So, the confidence here implies the likelihood that a river is multithread or single thread for a given land pixel. 0% confidence means it is neither type of river. Figure 3 made the concept easy to understand, but I suggest improving the writing here to make it clearer.

A2 – This may not have been clear in the original manuscript text and has now been improved. The influence of the confidence level on the river channel character is indeed best illustrated in Figures 3 and/or 6. However, we have clarified how to interpret the confidence in the introduction as well to provide more context throughout the manuscript. As the reviewer rightly points out, anything above a 0% confidence would indicate the likelihood that a land pixel is a type of river, whereas 0% would be a background (non-fluvial) value.

Q3 - L81 try “as well as.”

A3 – Implemented

Q4 - L126 cold (<-20 °C), missing a negative sign, same is L130. Please also check the others, like C.MI

A4 – Fixed and other values have been double checked.

Q5 - L196, I would add “(e.g., braided)” and “(e.g., meandering)”

A5 - Implemented